# Theobromine Improves Working Memory by Activating the CaMKII/CREB/BDNF Pathway in Rats

**DOI:** 10.3390/nu11040888

**Published:** 2019-04-20

**Authors:** Rafiad Islam, Kentaro Matsuzaki, Eri Sumiyoshi, Md Emon Hossain, Michio Hashimoto, Masanori Katakura, Naotoshi Sugimoto, Osamu Shido

**Affiliations:** 1Department of Environmental Physiology, Faculty of Medicine, Shimane University, 89-1 Enya-cho, Izumo 693-8501, Japan; rafiad@med.shimane-u.ac.jp (R.I.); erisumi@med.shimane-u.ac.jp (E.S.); emon@uab.edu (M.E.H.); michio1@med.shimane-u.ac.jp (M.H.); mkatakur@josai.ac.jp (M.K.); o-shido@med.shimane-u.ac.jp (O.S.); 2Department of Biotechnology and Genetic Engineering, Mawlana Bhashani Science and Technology University, Tangail 1902, Bangladesh; 3Department of Biochemistry and Molecular Genetics, University of Alabama at Birmingham, Birmingham AL 35294, USA; 4Department of Nutritional Physiology, Faculty of Pharmaceutical Sciences, Josai University, 1-1 Keyakidai, Sakado, Saitama 350-0295, Japan; 5Department of Physiology, Graduate School of Medical Science, Kanazawa University, 13-1 Takara-machi, Kanazawa 920-8640, Japan

**Keywords:** theobromine, cacao, working memory, behavior, CaMKII, CREB, BDNF

## Abstract

Theobromine (TB) is a primary methylxanthine found in cacao beans. cAMP-response element-binding protein (CREB) is a transcription factor, which is involved in different brain processes that bring about cellular changes in response to discrete sets of instructions, including the induction of brain-derived neurotropic factor (BDNF). Ca^2+^/calmodulin-dependent protein kinase II (CaMKII) has been strongly implicated in the memory formation of different species as a key regulator of gene expression. Here we investigated whether TB acts on the CaMKII/CREB/BDNF pathway in a way that might improve the cognitive and learning function in rats. Male Wistar rats (5 weeks old) were divided into two groups. For 73 days, the control rats (CN rats) were fed a normal diet, while the TB-fed rats (TB rats) received the same food, but with a 0.05% TB supplement. To assess the effects of TB on cognitive and learning ability in rats: The radial arm maze task, novel object recognition test, and Y-maze test were used. Then, the brain was removed and the medial prefrontal cortex (mPFC) was isolated for Western Blot, real-time PCR and enzyme-linked immunosorbent assay. Phosphorylated CaMKII (p-CaMKII), phosphorylated CREB (p-CREB), and BDNF level in the mPFC were measured. In all the behavior tests, working memory seemed to be improved by TB ingestion. In addition, p-CaMKII and p-CREB levels were significantly elevated in the mPFC of TB rats in comparison to those of CN rats. We also found that cortical BDNF protein and mRNA levels in TB rats were significantly greater than those in CN rats. These results suggest that orally supplemented TB upregulates the CaMKII/CREB/BDNF pathway in the mPFC, which may then improve working memory in rats.

## 1. Introduction

Coffee, cocoa, and chocolate are among the most frequently consumed substances in the world [1]. Coffee has various beneficial effects on human health, as it appears to be cardio-protective, neuroprotective, hepatoprotective, and nephroprotective. It is now well known that coffee (caffeine) suppresses the activities of nuclear factors κB (NF-κB), Akt, and ERK [2]. Like coffee, the use of the seeds from the cacao tree (Theobroma cacao) to create beverages, dates back to the early formative period of Mesoamerican history (2000–1000 BC). In recent years, there has been a notable interest in the neuroprotective effects of flavonoids, with evidence emerging that they may lead to improvements in memory and learning by improving neuronal functioning, while also promoting neuronal protection and regeneration [3]. Cacao beans, a very popular food worldwide, contains many flavonoids, which have pleotropic effects in cognition and neuroprotection [4,5]. Theobromine (TB) is a primary methylxanthine found in products made from cacao beans, which generally contain approximately 1% TB [6]. From recent literature, both scientific and popular, TB has been implicated in the health benefits of cacao intake. We know that TB traverses the blood-brain barrier (BBB) [7], and that this might induce effects on the brain, alter the cellular redox environment, modulate neuronal signaling pathways, and influence gene expression, as well as protein activity, perhaps in a manner similar to other flavonoids [8].

In our previous study, we confirmed that mice fed TB performed better on learning tasks and TB acted as a phosphodiesterase (PDE) inhibitor [9,10], by enhancing the cAMP/cAMP-response element-binding protein (CREB)/brain-derived neurotrophic factor (BDNF) pathway [7]. We also confirmed that TB supplemented chow-inhibited mTOR signaling in the brain and liver [11]. In the current experiment, we want to assess the effects of TB on rats, using a prolonged feeding period of 73 days, and to check whether different types of memory functions are affected by a variety of behavior experiments. We have checked the rats using the radial arm maze (RAM) task, Y-maze test, and novel object recognition (NOR) test for assessing the memory function, especially working memory. Working memory is the cognitive capacity to actively and temporarily maintain information for the purpose of task execution [12]. The dorsolateral prefrontal cortex in primates, which is homologous to the medial PFC (mPFC) in rodents [13,14], is essential for working memory, as evidenced by numerous lesion studies [15], electrophysiological recordings [16], and brain imaging investigations [17,18]. Many signaling pathways control the memory process, such as (i) cAMP-dependent protein kinase (PKA), (ii) Ca^2+^/calmodulin-dependent protein kinases (CaMKs), (iii) mitogen-activated protein kinases. These pathways converge to signal to CREB, a transcription factor, which is associated with memory and synaptic plasticity that binds to the promoter regions of many genes [19,20,21]. Remarkably, many genes have been altered following CREB activation [22,23], including key proteins involved in neuronal plasticity, such as BDNF [24]. This protein mediates neuronal development and synaptic function [24], which is critical for the differentiation and survival of neurons during development [25]. Studies have shown that Ca^2+^, acting as an important messenger via CaMKs, triggers phosphorylation of CREB [26]. This phosphorylated CREB (p-CREB) activates BDNF transcription by binding to a cAMP response element within the gene [24].

In this study, we found that TB fed rats appear to have improved working memories. As with the cAMP/CREB/BDNF pathway, which was established in our previous study for motor learning in mice, it appears that yet another novel molecular pathway, CaMKII/CREB/BDNF, may be responsible for working memory improvement in rats.

## 2. Materials and Methods

### 2.1. Animals

Forty four male Wistar rats (5 weeks old, 120~140 g body weight) purchased from Japan SLC Inc. (Shizuoka, Japan) were maintained at an ambient temperature of 24 ± 0.1 °C and relative humidity of 45% ± 5% under a 12:12-h light–dark cycle (light on at 7:00 h), with food and water ad libitum. All the animal experiments were performed in accordance with the Guidelines for Animal Experimentation of the Shimane University Faculty of Medicine, compiled from the Guidelines for Animal Experimentation of the Japanese Association for Laboratory Animal Science. The Committee on the Ethics of Animal Experiments of the Shimane University approved the protocol for this study.

### 2.2. Feeding and Experiment Schedules

Figure 1 and Figure 2 summarize the feeding and schedules for experiment 1 and 2. All of the rats had free access to a standard chow (CRF-1, Oriental Yeast Co. Ltd., Tokyo, Japan) for 10 days after admission. On day 0, the rats were divided into two groups. In both experiments, the control rats (CN rats), was fed the CRF-1 chow and the second group (TB rats), was fed the standard CRF-1 chow supplemented with 0.05% (W/W) of TB (Oriental Yeast Co. Ltd.) over the entire experiment time. 

#### 2.2.1. Behavioral Tests

##### RAM Task (Experiment 1)

CN rats (*n* = 12) and TB rats (*n* = 12) were behaviorally tested for their learning-related cognitive abilities by determining their ability to complete a task in a RAM, as described previously [27,28]. Here we used an eight arm radial maze (Toyo Sangyo, Toyama, Japan) for RAM task. Four weeks after the start of TB administration, rats were transferred to a regimen of food deprivation to keep their body weight at 80–85% of their free feeding weight, and each rat was handled for 3 minutes every day for a total of 5 consecutive days with constant monitoring of body weight. Their mean body weight was approximately 290 g at the beginning of the behavioral testing. Then, they were familiarized with the radial maze apparatus, across the entire surface on which reward pellets (Dustless Precision Pellets, Bio Serv^®^, Flemington, NJ, USA) were scattered. Then, the rats were trained to acquire a reward at the end of each of four arms of the eight arm radial maze. Three parameters of memory function were examined: Reference memory error (RME), which was determined by the number of entries into unbaited arms, working memory error (WME), which was estimated by the number of repeated entries into arms that had already been visited within a trial, and latency, which was determined by total time needed to finish each trial. A lower number of RMEs and WMEs suggested better spatial learning ability.

After the RAM task, the rats were anaesthetized using isoflurane and brains were rapidly separated from the skull, and the hippocampus were bilaterally collected. Afterwards, olfactory bulbs were removed and a coronal section was made on ice at +4.70 to +2.20 mm from bregma, according to a brain atlas [29]. The mPFC, containing the prelimbic, infralimbic, and anterior cingulate cortices, was immediately dissected from a coronal section. mPFC and hippocampal samples for Enzyme-linked immunosorbent assay (ELISA) were immediately frozen on liquid nitrogen and stored at −80 °C until use.

##### Y-Maze Test and NOR Test (Experiment 2)

It was clear that the procedural memory requirements and the stresses (both behavioral and metabolic) related to experimental procedures, including food deprivation, may non-specifically affect animal performance in the eight arm radial maze. Thus, to further assess the specificity of working memory improvement related to TB supplementation, we chose to test spontaneous alternation in the Y-maze test, which is devoid of all these procedural aspects, as it is based on the natural tendency of rats to explore novel environments. Another behavior test, the NOR test, in which spontaneous behavior is studied, where no artificial stimuli, food deprivation, reinforcement, and/or prior special training are required.

CN rats (*n* = 10) and TB rats (*n* = 10) were used for the Y-maze test and NOR test. Body weight was measured three times in the experimental time period. The Y-maze was performed twice: On day 48 and day 72. The NOR test was also performed twice, over days 52 to 54 and again from day 69 to day 71. All the experiments were done during the light phase from 10 a.m. to 6 p.m. Generally, the rats were placed in the experimental room at least 1 h prior to any test. In between subjects being tested, the apparatus was thoroughly cleaned with water wetted paper towels and a 70% ethanol solution.

Behavioral assessments were performed on rats to determine a spontaneous alteration of behavior using a Y-maze test, as this behavior is considered to reflect the strength of short-term memory [30,31,32]. The Y-maze was made of gray plastic with three arms (40 cm × 3 cm × 10 cm) extending from a central platform at an angle of 120°. Each rat was placed in the center of the arms and allowed to move freely around the three arms of the maze during an 8 min session. Arm entry was defined as the entry of four paws into one arm [33,34]. Alteration was defined over multiple sets. The percentage of spontaneous alternations were calculated as the ratio of the actual to possible alternations (defined as the total number of arm entries minus 2) multiplied by 100, as shown in the following equation: Alteration (%) = [(number of alternations)/(total arm entries-2)] × 100 [35]. For example, if the following sequence of arm entries was observed: ABACBCAABABC, the animal would have exhibited twelve arm entries, and three correct spontaneous alternations. The alteration percent here would be 40%. The behavior was recorded by a video camera located above the Y-maze, and correct spontaneous alterations were calculated at a later time by a trained observer.

Training and testing in the NOR task were carried out in an arena (70 × 70 × 30 cm) built of plywood as described by Ennaceur and Delacour [36]. The objects were made of plastic and were chosen after determining, in preliminary experiments with other animals, that they were equally preferred. Shapes, colors, and textures were different among these objects. Exploration of the objects was defined as sniffing or touching with the nose toward the objects at a distance of less than 1 cm; however, sitting on the object was not considered [37]. The circumstances where the rats explored the objects for <4 s were excluded. The animals were handled daily for the week that preceded the testing. Then, the animals were habituated to the NOR apparatus by placing them in it for 10 min per day to freely explore 1 day before the training (the habituation phase). On the training day, two identical objects (A1 and A2) were placed in the apparatus and the animals were allowed to explore them freely for 5 min (the familiarization phase). At one hour and 24 h later, in the test phase, one of the objects was randomly replaced by a novel object (named B, and C, respectively) and the rats were reintroduced into the apparatus for an additional 5 min period of free exploration [38,39,40]. To avoid confounds by lingering olfactory stimuli and preferences, the objects and the arena were cleaned with 70% ethanol after testing each animal. The Discrimination Index (DI), being the ability to distinguish the novel from the familiar object, was calculated: [novel object(s)/(novel object(s) + familiar object(s)) × 100%]. Thus, an index > 50% indicates novel object preference, <50% reveals a familiar object preference, and 50% implies no preference [41]. 

After the Y-maze test and NOR test, the rats were anaesthetized using isoflurane and the mPFCs, hippocampus, and plasma were sampled as described above. mPFC and hippocampal samples for ELISA were immediately frozen on liquid nitrogen and stored at −80 °C until use. mPFCs samples for western blot were immediately frozen on liquid nitrogen and stored at −80 °C until use. mPFCs samples for real time PCR were collected in RNAlatter reagent (Applied Biosystems, Warrington, UK), and stored at −30 °C until use. Whereas, plasma was used for liver and kidney function test as described below.

### 2.3. Liver and Kidney Function Test

Plasma was tested for the quantitative determination of ten parameters: glutamic oxaloacetic transaminase/aspartate transaminase (GOT/AST), glutamate–pyruvate transaminase/alanine transaminase (GPT/ALT), γ-glutamyl transpeptidase (GGT), triglyceride (TG), total cholesterol (T-Cho) and creatinine (Cre-2), albumin (Alb), total protein (T-Pro), urea acid (UA), blood urea nitrogen (BUN). These liver and kidney function tests were carried out using the automated biochemical analyzer Spotchem EZ SP-4430 (Arkray, Kyoto, Japan), and the Spotchem EZ Reagent Strip KENSHIN-2 (Arkray, Kyoto, Japan), Spotchem EZ Reagent Strip Kidney-3 (Arkray, Kyoto, Japan) were used. 

### 2.4. ELISA

The mPFCs were homogenized from rats of experiment 1 and 2 with Tris-buffer (pH 7.4) and centrifuged at 800× *g* for 15 min at 4 °C to remove tissue debris. Protein assays were performed using the Pierce BCA Protein Assay Kit (Thermo Fisher Scientific, Waltham MA, USA) to determine protein concentration. Homogenized samples were analyzed by ELISA, as described previously [42]. Briefly, equal amount of protein were analyzed with the BDNF Emax^®^ ImmunoAssay System (Promega, WI, USA) according to the manufacturer’s protocol. Absorbance, at 450 nm, was measured by a plate reader (DTX880, Beckman Coulter, CA, USA) and BDNF concentrations were calculated using SoftMax pro software (Molecular Devices, LLC, San Jose, CA, USA).

### 2.5. Western Blot Analysis

The mPFCs were extracted from rats of experiment 2 with a lysis buffer composed of 1 mM EDTA, 1% SDS, 1x complete protease inhibitor cocktail (Roche Diagnostics, Schweiz), 1x phosphatase inhibitor cocktail (Fujifilm Wako Pure Chemical Corporation, Osaka, Japan) and 20 mM Tris-HCl (pH 7.4). The lysates were sonicated and centrifuged at 14,000 rpm for 20 min at 4 °C to obtain the supernatant as the cell extract. Then, the lysates were analyzed by Western blotting as described previously [43,44,45]. Briefly, lysates were separated using a 10–12.5% SDS-PAGE system and transferred onto PVDF membranes (Immobilon-P, Merck Millipore, Burlington, MA, USA). Then, membranes were incubated with antibodies of monoclonal rabbit anti-p-CREB (Ser133) (1:1000, Cell signaling, Danvers, MA, USA), monoclonal mouse anti-CREB (1:1000, Cell signaling, Danvers, MA, USA), monoclonal rabbit anti-p-CaMKII (Thr^286/^Thr^287^) (1:1000, Cayman Chemical, MI, USA) and polyclonal rabbit anti-CaMKII beta (1:1000, GeneTex, CA, USA). HRP-conjugated anti-rabbit IgG (1:2000, Cell Signaling, Danvers, MA, USA) and HRP-conjugated anti-mouse IgG (1:2000, Cell Signaling, Danvers, MA, USA) were used as the secondary antibody. Immunoblots were incubated and visualized with the ECL detection kit (Amersham ECL Prime, GE Healthcare, Little Chalfont, Buckinghamshire, UK) and visualized with an image analyzer (LAS-4000, FUJI FILM, Tokyo, Japan). Then, membranes were stripped and reprobed with monoclonal rabbit anti-β-actin antibody as a loading control (1:2000, Cell Signaling, Danvers, MA, USA).

### 2.6. Real Time PCR

The total RNA from the mPFC was purified using a NucleoSpin® RNA isolation kit (TAKARA, Shiga, Japan) and reverse transcribed using a reverse transcription kit (TAKARA, Shiga, Japan). Real time PCR was carried out, as described previously, using the QuantiTect SYBR Green PCR Kit (Qiagen, Hilden, Germany) [46]. The quantification of mRNA was calculated using a cDNA sample as a calibrator. The quantified value of each sample was normalized with that of the β-actin value of the same sample, which was amplified simultaneously with the target gene. Appendix A shows the list of primer sequences used for real time PCR.

The PCR conditions were as follows: initial activation at 95 °C for 10 min, then 40 amplification cycles of denaturation at 95 °C for 15 s, followed by annealing and extension at 60 °C for 1 min.

### 2.7. Sample Size of Each Tested Assay

In RAM task (Experiment 1), 12 animals were used per group. In Y-maze (Experiment 2), 8 animals and in NOR test (Experiment 2), 5 animals were used. For ELISA of BDNF (Experiment 1), we used 5 animal from each group. For real time PCR, ELISA of BDNF, western blot analysis of p-CERB and p-CaMKII (Experiment 2), we used 6 animal per group.

### 2.8. Statistical Analysis

The data are expressed as the mean ± S.E.M. The Stat View statistical package (Version 5.0) was used for statistical analysis. A two-way analysis of variance (ANOVA) with Fisher’s PLSD post-hoc tests to determine significant differences in the various pairwise comparisons were used to analyze the RAM task results. Students’ *t*-tests were used to examine the differences between the two experimental groups in the Y-maze test, NOR test and biological analyses. The probability level of *p* < 0.05 was considered as statistically significant. 

## 3. Results

### 3.1. Experiment 1 (RAM Task)

#### 3.1.1. Body Weight

First, we measured the body weights at three different time point during the experiment. The body weights of TB rats (*n* = 12) did not differ from those of CN rats (*n* = 12) on day 0 (*p* = 0.32), day 45 (*p* = 0.29) or day 73 (*p* = 0.12) (Table 1).

#### 3.1.2. Effects of TB on Memory Related Learning Abilities on the RAM Task

The effects of TB supplementation on reference and working memory-related learning abilities were determined by examining the changes in the mean number of WMEs and RMEs, with the data averaged over a block of five trials (Figure 3A,B). Randomized two factor (block and group) ANOVA was used to analyze the possible impact of TB, and revealed a significant main effect in both blocks of trials [F (4,88) = 17.583, *p* < 0.0001] and groups [F (1,88) = 27.265, *p* < 0.0001] on the number of WMEs, with a significant block x group interaction [F (4,88) = 3.131, *p* < 0.05]. However, even without a significant main group effect, and block x group interaction on the number of RMEs, a significant main effect of blocks of trials [F (4,88) = 11.774, *p* < 0.0001)] were observed. Another metric, latency was also determined by examining changes in the total mean time, with the data averaged over a block of five trials (Figure 3C). We found a significant main effect in both the blocks of trials [F (4,88) = 27.882, *p* < 0.0001] and the groups [F (1,88) = 21.558, *p* < 0.0001] on the number of total times, with a significant block x group interaction [F (4,88) = 2.815, *p* < 0.05]. 

#### 3.1.3. BDNF Protein Expression Level in the mPFC

BDNF protein expression level in the mPFC of TB rats (*n* = 5) was significantly higher than that of CN rats (*n* = 5) (Figure 4; *p* = 0.009).

### 3.2. Experiment 2 (Y-Maze Test and NOR Test)

#### 3.2.1. Body Weight, Liver Function, and Kidney Function Tests

The body weights of TB rats (*n* = 10) did not differ from those of CN rats (*n* = 10) on day 0 (*p* = 0.47), day 45 (*p* = 0.30), or day 73 (*p* = 0.21), respectively (Table 2). The biochemical parameters for liver and kidney functions of TB rats (*n* = 10) also did not differ from those of CN rats (*n* = 10) rats after day 73 (Table 3). These results indicated that TB did not affect the feeding behavior and normal liver or kidney functions.

#### 3.2.2. Effects of TB Supplementation on Working Memory Improvement in Y-Maze Test

The exploration rates were not affected by TB supplementation, as the total number of arm entries between groups did not show any significant difference (Figure 5A). TB rats displayed spontaneous alternations of 68.9 ± 1.72 and 72.5 ± 2.41 of choices, which was significantly higher than the 60.2 ± 2.59 and 57.8 ± 3.04 of values for the control group (Figure 5B, *p* = 0.014, *p* = 0.002), as measured across two trials, respectively.

#### 3.2.3. Effects of TB Supplementation on Memory Function in the NOR Test

During the training session (the familiarization phase), the rats spent a similar period of time exploring two identical objects. Figure 6A shows the DIs for the identical objects (A1 and A2) between the two groups were statistically not significant in both trials (*p* = 0.478 for trial 1, *p* = 0.627 for trial 2). 

Figure 6B shows that the DIs for object A1 and object B (after 1 h object A2 was replaced by object B) in test phase was significantly increased between groups in both trials (*p* = 0.043 for trial 1, *p* = 0.005 for trial 2).

In Figure 6C, the DIs for object A1 and novel object C (after 24 h object B was changed to new object C) were significantly higher in TB rats in both trials (*p* = 0.041 for trial 1, *p* = 0.012 for trial 2). The results suggest that TB rats has a > 50% DI, implying a higher novel object preference. 

To help determine whether explorative and/or anxiety-like behaviors are affected, it is quite important to assess the total exploration time spent at an object. Total exploration time for the same object did not show significant differences between groups in both trials (Figure 6D, *p* = 0.744 for trial 1 and *p* = 0.509 for trial 2), whereas the total exploration times regarding objects A1 and B (Figure 6E, *p* = 0.006 for 1st trial and *p* = 0.037 for 2nd trial), and objects A1 and C (Figure 6F, *p* = 0.036 for trial 1 and *p* = 0.005 for trial 2) were significantly different between groups in both trials. However, the CN rats did not show that much of preference to novel object. Though, the average time spent near the novel object increased numerically in the control rats, however, it did not reach significance.

#### 3.2.4. p-CaMKII, p-CREB and BDNF mRNA and Protein Expression in the mPFC

The expression level of p-CaMKII in the mPFC was significantly higher in TB rats than that of CN rats (Figure 7A, *p* = 0.005). The expression level of p-CREB in the mPFC was also significantly higher in TB rats than that of CN rats (Figure 7B, *p* = 0.023). BDNF mRNA and protein expression levels in the mPFC were also significantly higher in TB rats than in their CN counterparts (Figure 8A, *p* = 0.030 and Figure 8B, *p* = 0.015). 

## 4. Discussion

The results of our present study demonstrated that the oral administration of TB influenced the signaling pathway in the mPFC, including those for CaMKII, CREB and BDNF, and concurrently improved the working memory function of rats in the RAM task, the Y-maze test, and also in the NOR test. Moreover, long-term TB supplementation did not show any adverse effects on body weight, food, water intake, liver, and kidney functions in rats (Table 1, Table 2 and Table 3).

To exert material effects against cognitive disorders, TB must be taken up in the brain from the blood by crossing over the BBB. Our previous data showed that TB was detectable in rat plasma and the brain after 30 and 40 days of TB ingestion, and gradually increased in a time-dependent manner [11]. The concentration of TB in rat brain and plasma after 40 days of TB ingestion might have been sufficient to produce significant pharmacological effects, as described in [4,7,47].

Previously we showed that a 30-day orally administrated TB enhances motor learning in mice [7]. In the present study, we prolonged TB administration to 73 days, and found it intriguing that the TB fed group of rats made fewer WMEs and lower latency in the RAM task, as the number of trials increased. In order to have the food efficiently with minimal effort when a baited arm is visited for food, the rat has to avoid re-entry, and this learning strategy involves working memory. The results of the decreased WMEs and shorter latency suggest that the administration of TB significantly improved the short-term/working memory (Figure 3A). However, in our experimental setting, it could not affect the long-term memory, as RME scores between the two groups were not significantly different (Figure 3B). Concurrently, TB, compared to control animals, produced the learning-related memory significantly at a faster pace (shorter time) (Figure 3C). Oral administration of TB also triggered a significant increase in the spatial working memory, as indicated by a higher spontaneous alteration ratio in the Y-maze test [48]. TB not only ameliorated the RAM task- and Y-maze-determined working memory, but also significantly contributed to the enhancement of memory examined in NOR test, which was determined with an Inter Trial Interval (ITI) at 1h, and 24h, respectively. This was confirmed by increases in the DI and exploration time at both 1h and 24h of the post-familiarization test, which respectively can be referred to as short-term and long-term memory. With regards to this, Ennaceur et al. (1997) and Quillfeldt et al. (2016) also reported that object recognition in the NOR task is usually more employed to examine working memory [49,50], while recognition tasks tested the ‘delays’ of more than 6h allow one to determine the long-term memory [49,50]. Therefore, TB supplementation is significant in the betterment of learning-related memory cognition. 

Many mechanisms have been proposed in relation to the location of memory. The working memory, which relates to the ‘temporary operation and storage of information’, is mainly stored in the prefrontal cortex of the brain [12,51]. Albeit, there are some reports that working memory is also stored in the hippocampus, at least to some extent [51]. On the other hand, long-term memory formation mainly occurs in the hippocampus [52,53]. However, the neocortex [54] and perirhinal cortex [55,56] of the medial temporal lobe are anatomically interlinked in the formation of long-term memory [57]. The long-term memory could be better described to occur in both the hippocampus and cortex regions, while the former is used for the formation and/or storage of new memory trace, and the latter is needed for long term storage [58,59,60,61,62]. Thus suggesting that the exact location of the neural circuitry of the memory is yet to be clearly elucidated.

The enhancement of both working- and long-term memory is controlled at the molecular level in neurons [63]. Whereas, working memory involves modifications of pre-existing neurochemicals/proteins, and long-term memory requires the synthesis of new mRNAs and proteins [64,65]. Furthermore, memory encompasses cholinergic, noradrenergic, dopaminergic systems [66], and most importantly glutamatergic system is critical to the learning and memory, and related plasticity [67,68]. *N*-methyl-d-aspartate receptor (NMDAR) is a voltage sensitive glutamate receptor. Glutamate-NMDAR interaction causes activation of Ca^2+^-calmodulin cascade, including CaMKII [69]. Functional maintenance of neuronal circuitry depends on different neurotrophic factors like nerve growth factor (NGF), glial cell line-derived neurotrophic factor (GDNF), BDNF, etc. [70,71,72]. Besides the BDNF, many other signaling proteins e.g., CaMKII, MAPK, PKC, PKA, PI3K/Akt [73,74,75,76,77], and transcription factors, such as CREB [78], are important in memory-related neural plasticity. For example, CaMKII is a key synaptic signaling molecule that facilitates learning and memory processes by mediating a wide variety of intercellular signals [79,80]. CaMKII contributes to long term potentiation (LTP) and hence long-term memory [81]. 

Many studies have suggested that CaMKII phosphorylates the transcription factor CREB and transforms it into active form p-CREB [73,75,77]. p-CREB then initiates transcription and translation of proteins/receptors required for neuronal plasticity. This could be confirmed by inhibiting p-CREB and subsequent inhibition of new protein synthesis and, therefore, impair memory cognition [75]. In the current study, TB supplementation significantly upregulated CaMKII, as indicated by increased ratios of p-CaMKII/CaMKII in the cortical tissues (Figure 7A). TB augmented the levels of p-CREB concomitantly, as compared to those of the control rats (Figure 7B). Therefore, we speculate that TB-induced an increase in the levels of p-CaMKII, and p-CREB contributed to the improvement of neuronal plasticity, hence, the learning and memory of TB rats. BDNF has been implicated in LTP that occurs in the hippocampus and other brain regions. LTP plays a prime role in learning and memory [82,83]. Improvements of working memory were accompanied with an increased level of BDNF proteins in the frontal cortex [84,85]. Consistently, TB-fed rats, in the current study, had also higher BDNF levels in the mPFC. Moreover, the improvements in learning and memory were positively correlated with the levels of memory-related substrates—p-CaMKII, p-CREB, and BDNF (Appendix A). Therefore, it is conceivable that TB-instigated increases in the levels of p-CaMKII, p-CREB, and BDNF improved the learning and the memory of the rats. TB supplementation did not have an effect on ERK1/2 pathway, as indicated by no change in the level of p-c-Raf, p-MEK1/2, p-90RSK; and p-MSK1 in the mPFC (Appendix A). However, the cause remains to be clarified. 

There are accumulating shreds of evidences that the deficit in working memory has been implicated in several neurodegenerative (Parkinson’s disease, Alzheimer’s disease, and ageing) and/or neurodevelopmental disorders, such as attention deficit hyperactivity disorder (ADHD), Schizophrenia [86,87,88,89]. ADHD is associated with loss of function of the mPFC, which plays a role in regulating complex cognitive, emotional, and behavioral activities [90]. Interestingly, Yabuki et al. reported that deficits in working memory in ADHD model rats and spontaneously hypertensive rats (SHR), were closely associated with dysfunction of CaMKII in the mPFC, but not in the hippocampus [91]. Since the accumulation of TB in the cortex has been clearly detected [7,11], neurodegenerative and/or neurodevelopmental disorders, associated with these regions, might be beneficent from the oral administration of TB. Moreover, this study demonstrated that chronic oral administration of TB did not cause any adverse health effects in rats. TB may be considered as a safe functional food to include in the daily diet.

TB can be metabolized from caffeine, and both caffeine and TB may work through a similar pathway. Caffeine has some pharmacological effects in the central nervous system and beneficial effects on memory and/or cognitive functions in humans and rodents [92,93,94,95,96]. Since TB can be metabolized from caffeine, these components may work through similar mechanisms. For instance, caffeine and TB can pass through BBB and block cell surface adenosine receptors, which distributed widely throughout cortical regions [97]. TB and caffeine also act as PDE inhibitor that increases in intracellular cAMP level [7,11,97]. Moreover, these reagents are known to affect Ca^2+^ release from intracellular stores of the brain [97]. Although, exact differences between caffeine and TB on memory and/or cognitive functions have not been examined in this study, a comparison study for these reagents may be required in the future.

## 5. Conclusions

This study demonstrated that the oral administration of TB for 73 days resulted in the upregulation of p-CaMKII and p-CREB in the mPFC, and also found both BDNF mRNA expression level, as well as protein level upregulation in the mPFC of TB rats. These results clearly suggest that TB supplementation may facilitate the CaMKII/CREB/BDNF pathway in the mPFC. We also clearly observed a significant improvement in working memory in TB rats. These observations are also firmly supported by previous findings concerning the role of the CaMKII/CREB/BDNF pathway in working memory and learning in rats.

## Figures and Tables

**Figure 1 nutrients-11-00888-f001:**
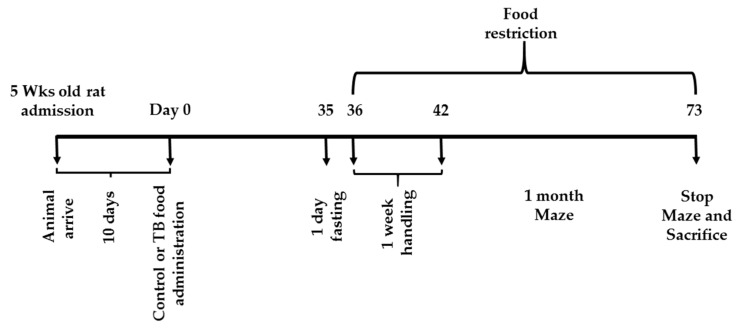
Schedule for Experiment 1 (radial arm maze (RAM) task).

**Figure 2 nutrients-11-00888-f002:**
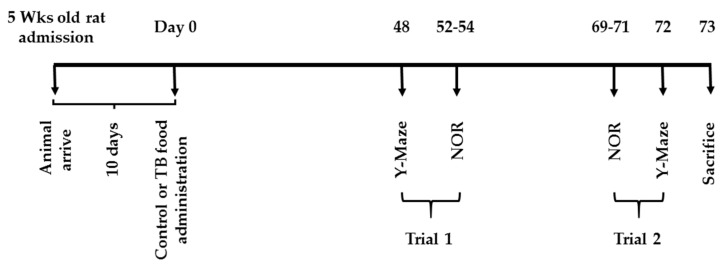
Schedule for Experiment 2 (Y-maze test and novel object recognition (NOR) test).

**Figure 3 nutrients-11-00888-f003:**
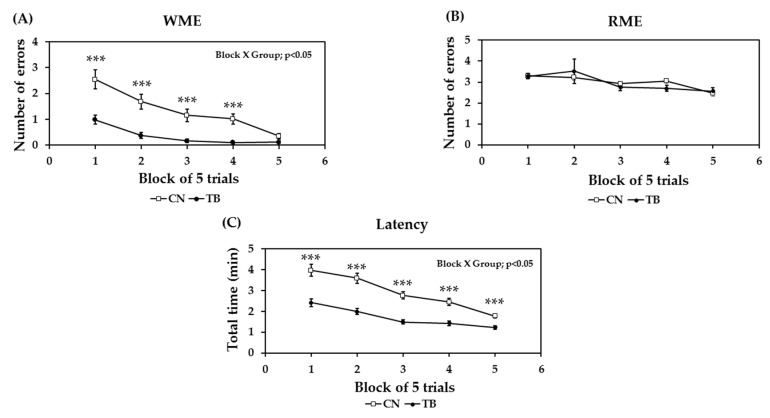
The effect of theobromine (TB) on (A) working memory errors (WME); (B) reference memory errors (RME); and (C) latency assessed by the radial arm maze task. Results are expressed as the mean ± S.E.M. (*n* = 12 for each group). *** *p* < 0.001; significant difference between groups.

**Figure 4 nutrients-11-00888-f004:**
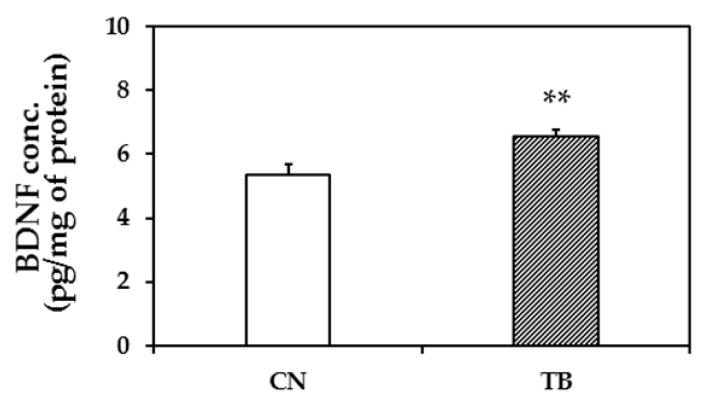
The effect of theobromine (TB) on the brain-derived neurotrophic factor (BDNF) protein level in the medial prefrontal cortex (mPFC). BDNF protein level in the mPFC of TB-fed rats (TB rats) was significantly higher than that of control rats (CN rats). Hippocampal BDNF protein levels did not differ between groups (Appendix A). Values are the mean ± S.E.M. (*n* = 5 for each group). ** *p* < 0.01; significant difference between groups.

**Figure 5 nutrients-11-00888-f005:**
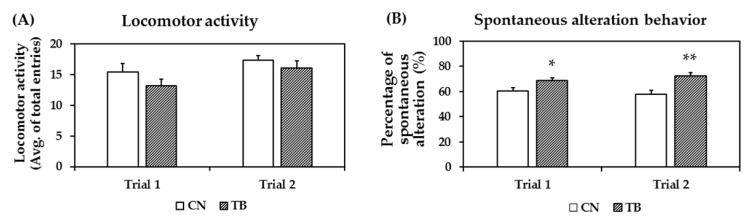
The effect of theobromine (TB) on locomotor activity and spontaneous alteration behavior in rats assessed by the Y-maze test. (**A**) Locomotor activity of TB-fed rats (TB rats) and control rats (CN rats). There were no significant differences between groups in both trials. (**B**) TB rats had a significantly higher number of alteration in both trials compared to CN rats. Values are the mean ± S.E.M. (*n* = 8 for each group). * *p* < 0.05, ** *p* < 0.01; significant difference between groups.

**Figure 6 nutrients-11-00888-f006:**
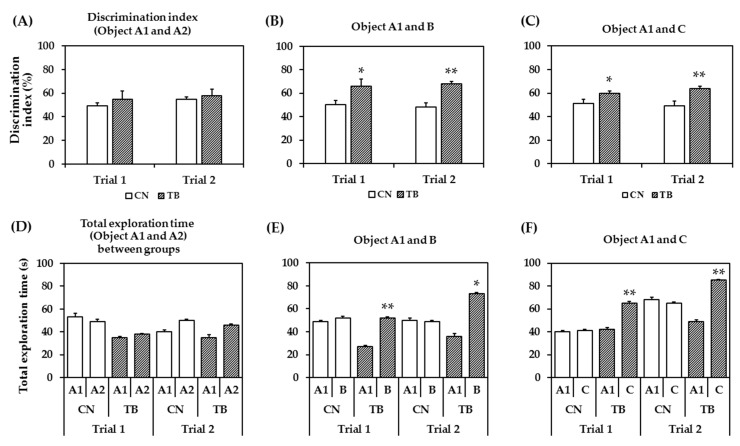
The effect of theobromine (TB) on memory performance assessed by the novel object recognition (NOR) test. The discrimination index (DI) for exploring the object A1, A2 (**A**); object A1, B (**B**); and object A1, C (**C**). Both DIs (Object A1, B and Object A1, C) were significantly greater in TB-fed rats (TB rats) in 2 trials. Values are the mean ± S.E.M. (*n* = 5 for each group). * *p* < 0.05, ** *p* < 0.01; significant difference between groups. Total exploration time for same object (object A1, A2) (**D**); total exploration time for each object (object A1, B) (**E**); and total exploration time for each object (object A1, C) (**F**). TB rats stayed significantly longer time at the new object B and C in 2 trials. Values are the mean ± S.E.M. (*n* = 5 for each group). * *p* < 0.05, ** *p* < 0.01; significant difference between A1 and B or C.

**Figure 7 nutrients-11-00888-f007:**
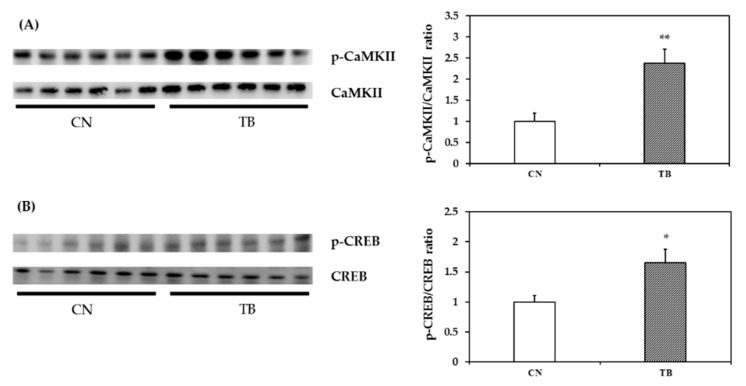
Phospho-CaMKII (p-CaMKII) and phospho-CREB (p-CREB) levels in the medial prefrontal cortex (mPFC) of control rats (CN rats) and theobromine (TB)-fed rats (TB rats). TB enhanced (**A**) p-CaMKII; and (**B**) p-CREB levels in the mPFC. Values are the mean ± S.E.M. (*n* = 6 per group). * *p* < 0.05, ** *p* < 0.01; significant difference between groups.

**Figure 8 nutrients-11-00888-f008:**
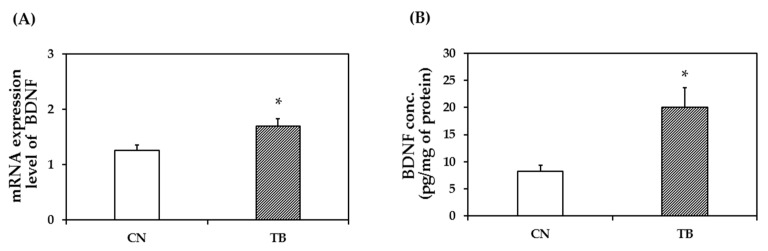
The effect of theobromine (TB) on brain derived neurotrophic factor (BDNF) mRNA and protein levels. (**A**) Relative mRNA expression and (**B**) protein level in TB-fed rat (TB rats) of BDNF were significantly higher than those of control rats (CN rats) in the prefrontal cortex. Hippocampal BDNF protein levels did not differ between groups (Appendix A). Values are the mean ± S.E.M. (*n* = 6 for each group). * *p* < 0.05; significant difference between groups.

**Table 1 nutrients-11-00888-t001:** Body weights for control rats (CN rats) and theobromine-fed rats (TB rats) in experiment 1.

	Body Weight (g)		*p* Value
	CN Rats	TB Rats	
Initial day (day 0)	239.2 ± 7.13	229.9 ± 5.20	0.32
Half way (day 45)	360.6 ± 3.95	353.6 ± 4.98	0.29
Final (day 73)	384.9 ± 3.10	375.5 ± 4.65	0.12

Values are the mean ± S.E.M. (*n* = 12 for each group).

**Table 2 nutrients-11-00888-t002:** Body weights for control rats (CN rats) and theobromine-fed rats (TB rats) in experiment 2.

	Body Weight (g)		*p* Value
	CN Rats	TB Rats	
Initial day (day 0)	183.1 ± 2.03	185.5 ± 2.52	0.47
Half way (day 45)	416.0 ± 5.67	428.3 ± 9.92	0.30
Final (day 73)	463.0 ± 7.89	481.0 ± 11.3	0.21

Values are the mean ± S.E.M. (*n* = 10 for each group).

**Table 3 nutrients-11-00888-t003:** Biochemical parameters for liver function and kidney function in experiment 2 for control rats (CN rats) and theobromine-fed rats (TB rats).

Liver Function Test				Kidney Function Test			
Biochemical Parameters	CN Rats	TB Rats	*p* Value	Biochemical Parameters	CN Rats	TB Rats	*p* Value
GOT/AST (IU/L)	67.4 ± 8.43	63.7 ± 8.14	0.76	T-Pro (g/dl)	5.4 ± 0.10	5.2 ± 0.12	0.09
GPT/ALT (IU/L)	25.1 ± 1.74	23.7 ± 2.23	0.63	Alb (g/dl)	3.1 ± 0.05	3.0 ± 0.07	0.50
GGT (IU/L)	2.7 ± 0.21	3.3 ± 0.21	0.06	BUN (mg/dl)	16.6 ± 0.40	16.3 ± 0.50	0.64
T-Cho (mg/dl)	79.0 ± 4.94	90.6 ± 5.74	0.14	UA (mg/dl)	1.2 ± 0.06	1.0 ± 0.09	0.11
TG (mg/dl)	95.0 ± 10.97	102.6 ± 13.44	0.67	Cre-2 (mg/dl)	0.3 ± 0.02	0.3 ± 0.02	0.75

GOT/AST, glutamic oxaloacetic transaminase/aspartate transaminase; GPT/ALT, glutamate–pyruvate transaminase/alanine transaminase; GGT, γ-glutamyl transpeptidase; T-Cho, total cholesterol; TG, triglyceride; T-Pro, total protein; Alb, albumin; BUN, blood urea nitrogen; UA, urea acid; Cre-2, creatinine. Values are the mean ± S.E.M. (*n* = 10 for each group).

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
