# Peer review of "Theobromine Improves Working Memory by Activating the CaMKII/CREB/BDNF Pathway in Rats"

_nutrients, 2019, doi:10.3390/nu11040888_

Round 1
Reviewer 1 Report
The manuscript by Islam et al. supports the capacity of dietary theobromine to facilitate the mnemonic performance of rats. In general, the manuscript is well-written and the subject matter is of interest to the present readership. However, there are many methodological concerns that require clarification as well as some concerns regarding statistical analysis and interpretation. Please, see details below:
1) Throughout the manuscript, it is not clear when the authors intend to make assertions about working memory, short-term memory, or long-term memory.
a. The manuscript appears to have conducted behavioral tasks that assess working or short-term memory as well as long-term memory, yet authors largely refer to all findings for their influence on “working memory.” Authors need to consider that an ITI of 1 h (working memory) and 24 h (long-term memory) recruit different brain regions and signaling pathways beyond BDNF actions in the PFC. As such, these findings should be properly referred to as either “working” or “long-term” memory and the discussion of each should include the potential involvement of hippocampal or additional cortical regions and signaling factors that may interact with those they have investigated.
b. Why were the Y-maze and NOR tests run twice (what is the benefit)? Were different objects used when the NOR test was run again? Exploration appears higher on the second trial which suggests that there was an effect of repeated testing.
2) It is of concern that the control rats did not appear to learn in the NOR test.
a. It is appreciated that TB increased the proportion of time spent with the novel object, but even control rats should show some preference for novelty. Can the test be considered valid if controls did not demonstrate learning even after a 1 h ITI?
b. The n/group is very low for this test (n=5). Increasing the number of observations may address the concern raised above.
3) Please, describe the gross brain dissection and indicate what region of PFC was collected (mPFC? The entire frontal cortex?) Including some indication as to boundaries that were used to make the dissection will allow inferences to be made regarding the subregions included (e.g. ACC, prelimbic, agranular insular, infralimbic cortex, etc).
4) There appears to be a great deal of variance in the expression of p-CREB (Fig. 7B). Did the expression of this or other markers correlate with any behavioral measure?
5) The radial arm maze should be analyzed as a repeated-measures ANOVA. As well, please indicate on Fig. 3A and 3C which datapoints significantly differ from control.
MINOR COMMENTS
6) The diagrams in Fig. 1 and 2 have some points of confusion and could be improved for clarity.
a. Please, indicate how much time passed from “animal arrival” when the animals were 35 days old and “Day 0” when the animals went into protocol.
b. The label “TB food administration” should be “Control or TB food administration”
c. The word “starvation” is presumed to be inaccurate and should be replaced with “fasting”
d. The Methods section indicates a week of handling prior to testing; this should be indicated on the diagram.
7) The data that is omitted in the instances of “data not shown,” should be included in a supplemental file. These data address points of interest to this reader.
8) While the manuscript is largely well-written, there are many instances of typographical errors. This is not a complete list:
a. Pg. 2, “transverses” should be “traverses”
b. Pg. 2, “medium” should be “medial”
c. Pg. 4 and throughout the manuscript, “alterations” should be “alternations”
d. Pg. 11, “trails” should be “trials”
Author Response
To Reviewer #1
(All revisions have been made in blue color in this response sheet.)
The manuscript by Islam et al. supports the capacity of dietary theobromine to facilitate the mnemonic performance of rats. In general, the manuscript is well-written and the subject matter is of interest to the present readership. However, there are many methodological concerns that require clarification as well as some concerns regarding statistical analysis and interpretation. Please, see details below:
Comment #1:
Throughout the manuscript, it is not clear when the authors intend to make assertions about working memory, short-term memory, or long-term memory.
1(a) The manuscript appears to have conducted behavioral tasks that assess working or short-term memory as well as long-term memory, yet authors largely refer to all findings for their influence on “working memory.” Authors need to consider that an ITI of 1 h (working memory) and 24 h (long-term memory) recruit different brain regions and signalling pathways beyond BDNF actions in the PFC. As such, these findings should be properly referred to as either “working” or “long-term” memory and the discussion of each should include the potential involvement of hippocampal or additional cortical regions and signalling factors that may interact with those they have investigated.
Answer:
Thanks for your observation. As we conducted different behavioural task to access memory function. In radial arm maze (RAM) task, working memory errors (WMEs) reflect short-term memory/working memory and Reference memory errors (RMEs) reflect long-term memory. In our RAM experiment, the working memory was improved whereas we did not find any improvement in long term memory in TB rats.
The Y-maze task is mainly used to examine working memory performance [1–3]. We found an ameliorating effect of TB on working memory in TB rats.
In Novel object recognition (NOR) test, 1 h after the familiarization task (i.e., at an ITI of 1h) the performance is considered as a measure of working memory. The performance measured at 24 h after the familiarization task (i.e., at an ITI of 24 h) is referred to long-term memory [4,5]. In NOR test TB ameliorated both working and long term memory. Unfortunately, we missed to mentioning the effect of TB on long term memory in the previous manuscript. However, we have mentioned it in the revised manuscript.
Regarding the location of memory in different brain regions:
Answer:
We agree with you that memory-related brain cognition is a complex interplay among different brain regions, primarily the hippocampus and cortex, in particular, medial prefrontal cortex (mPFC). Also, the prefrontal cortex, infralimbic cortex, prelimbic, etc. and even the thalamus are involved in memory-related brain cognition [6][7][8][9]. This suggests that no-specific region has yet been assigned for a given memory formation in general.
Working and Long-term memory:
Answer:
The working memory which relates to ‘temporary operation and storage of information’ is mainly stored in the prefrontal cortex of brain [10][11]. Albeit, there are some reports that working memory also stored in the hippocampus [10], at least, to some extent.
On the other hand, long-term memory formation mainly occurs in the hippocampus [12][13]. However, the neocortex [14] and perirhinal cortex [15][16] of the medial temporal lobe are anatomically interlinked in the formation of long-term memory [17]. The long-term memory is better described to occur in both hippocampus and cortex regions, while the former is used for the formation/storage of new memory trace, and the latter is needed for long term storage [6,18–21].
Regarding the location of different memory in the hippocampus vs. cortex, finally, we have added the following sentences in the revised manuscript:
Page#12
Line#382-391
Many mechanisms have been proposed on about the locations of memory. The working memory which relates to ‘temporary operation and storage of information’ is mainly stored in the prefrontal cortex of the brain [10,11]. Albeit, there are some reports that working memory is also stored in the hippocampus, at least, to some extent [10]. On the other hand, long-term memory formation mainly occurs in the hippocampus [12,13]. However, the neocortex [14] and perirhinal cortex [15,16] of the medial temporal lobe are anatomically interlinked in the formation of long-term memory [17]. The long- term memory could be better described to occur in both the hippocampus and cortex regions, while the former is used for the formation and/or storage of new memory trace, and the latter is needed for long term storage [6,18–21]. Thus suggesting that the exact location of the neural circuitry of the memory is yet to be clearly elucidated.
As suggested by the reviewer 1, we have also inserted the following sentences regarding the performance of NOR test measured at 1h ITI and 24h ITI, as working/short-term and long-term memory respectively:
Page# 12
Line#372-379
TB not only ameliorated the RAM task- and Y-maze-determined working memory, but also significantly contributed to the enhancement of memory examined in NOR test determined with an Inter Trial Interval (ITI) at 1h and 24h, respectively. This was confirmed by increases in the DI and exploration time at both 1h and 24h of the post-familiarization test, which respectively can be referred to as short-term and long-term memory. In regard to this, Ennaceur et al. (1997) and Quillfeldt et al. (2016) also reported that object recognition in the NOR task is usually more employed to examine working memory [4,5], while recognition tasks tested with ‘delays’ of more than 6h allow one to determine the long-term memory [4,5].
Signalling pathways beyond BDNF action:
Answer:
Thank you, we fully agree with you. Beside the action of BDNF, many other signalling proteins are involved. Therefore, we have added the following sentences as per your suggestion:
Page#12-13
Line#392-423
The enhancement of both working- and long-term memory is controlled at the molecular level in neurons [22]. Whereas working memory involves modifications of pre-existing neurochemicals/ proteins, long-term memory requires the synthesis of new mRNAs and proteins [23,24]. Furthermore, memory encompasses cholinergic, noradrenergic, dopaminergic systems [25], and most importantly glutamatergic system is critical to the learning and memory, and related plasticity [26,27]. N-methyl-D-aspartate receptor (NMDAR) is a voltage sensitive glutamate receptor. Glutamate-NMDAR interaction causes activation of Ca2+-calmodulin cascade, including CaMKII [28]. Functional maintenance of neuronal circuitry depends on different neurotrophic factors like nerve growth factor (NGF), glial cell line-derived neurotrophic factor (GDNF), BDNF, etc. [29–31]. Beside the BDNF, many other signaling proteins e.g., CaMKII, MAPK, PKC, PKA, PI3K/Akt [32–36], and transcription factors such as CREB [37], are important in memory-related neural plasticity. For example, CaMKII is a key synaptic signaling molecule that facilitates learning and memory processes by mediating a wide variety of intercellular signals [38,39]. CaMKII contributes to long term potentiation (LTP) and hence long-term memory(LTM) [40].
Many studies have suggested that CaMKII phosphorylates the transcription factor CREB and transforms it into active form p-CREB [32,34,36]. p-CREB then initiates transcription and translation of proteins/receptors required for neuronal plasticity. This could be confirmed by inhibiting p-CREB and subsequent inhibition of new protein synthesis and therefore impair memory cognition [75]. In the current study, TB supplementation significantly upregulated CaMKII, as indicated by increased ratios of p-CaMKII/CaMKII in the cortical tissues (Figure 7A). TB augmented the levels of p-CREB concomitantly, as compared to those of the control rats (Figure 7B). Therefore, we speculate that TB-induced increases in the levels of p-CaMKII and p-CREB contributed to the improvement of neuronal plasticity, hence, the learning and memory of TB-rats.
BDNF has been implicated in LTP that occurs in the hippocampus and other brain regions. LTP plays a prime role in learning and memory [41,42]. Improvements of working memory were accompanied with an increased level of BDNF proteins in the frontal cortex [43,44]. Consistently, TB-fed rats, in the current study, had also higher BDNF levels in the mPFC. Moreover, the improvements in learning and memory were positively correlated with the levels of memory-related substrates¾p-CaMKII, p-CREB, and BDNF (supplementary file 2). Therefore, it is conceivable that TB-instigated increases in the levels of p-CaMKII, p-CREB, and BDNF improved the learning and the memory of the rats. TB supplementation did not have an effect on ERK1/2 pathway, as indicated by no change in the level of p-c-Raf, p-MEK1/2, p-90RSK; and p-MSK1 in the mPFC (supplementary file 2). However, the cause remains to be clarified.
Comment #1(b). Why were the Y-maze and NOR tests run twice (what is the benefit)? Were different objects used when the NOR test was run again? Exploration appears higher on the second trial which suggests that there was an effect of repeated testing.
Comment: Why were the Y-maze and NOR tests run twice (what is the benefit)?
Answer:
The Y-maze and NOR test were run twice in two different time points (in the middle, and at the end of our experimental schedule). Our aim was to check whether the memory is improved to a greater extent in later time point. But, we found almost similar differences in the values of discrimination index in both trials.
Comment: Were different objects used when the NOR test was run again?
Answer:
In the second trial of NOR test, the same set of objects was used.
Comment: Exploration appears higher on the second trial which suggests that there was an effect of repeated testing.
Answer:
Thank you. We agree with you.
Comment #2:
It is of concern that the control rats did not appear to learn in the NOR test.
2(a). It is appreciated that TB increased the proportion of time spent with the novel object, but even control rats should show some preference for novelty. Can the test be considered valid if controls did not demonstrate learning even after a 1 h ITI?
Answer:
Thank you for raising the issue. In our experiment, we found that TB rats spent more time with novel object. However the control rats did not show that much of preference to novel object. Though, the average time spent near the novel object increased numerically in the control rats, however, it did not reach significance. This might be accredited to the influence of theobromine in the TB-rats. Otherwise, the exploration time could have not been increased in the TB rats.
Regarding this issue we have added two lines in the manuscript “results” section, are as below:
Page#9
Line#319-321
However, the control rats did not show that much of preference to novel object. Though, the average time spent near the novel object increased numerically in the control rats, however, it did not reach significance.
Comment #2:
2(b). The n/group is very low for this test (n=5). Increasing the number of observations may address the concern raised above.
Answer:
Thanks for your comment. To fulfil different criteria on NOR test we had to use 5-rats per group [45–49] for the behavioral analysis. We had to exclude 2-rats from the Y-maze test because they failed to have entered into the arms at least a total of 6 times. In the NOR test, again we had to exclude more 3-rats, as these 3-rats failed to explore the objects within the time limit of 4 seconds. However, we fully agree with you, if we could have more rats the results could be better. It would be highly appreciated if you kindly consider the issue.
Comment #3:
Please, describe the gross brain dissection and indicate what region of PFC was collected (mPFC? The entire frontal cortex?) Including some indication as to boundaries that were used to make the dissection will allow inferences to be made regarding the subregions included (e.g. ACC, prelimbic, agranular insular, infralimbic cortex, etc).
Answer:
Thank you for pointed out an important matter. In this study, medial prefrontal cortex (mPFC), containing the prelimbic, infralimbic, and anterior cingulate cortices, was used for molecular-biological assays. The mPFC and hippocampal samples were isolated as follows:
Following decapitation under anaesthesia, brains were rapidly separated from the skull, and then the hippocampus was bilaterally collected. Afterwards, olfactory bulbs were removed and the coronal section was made on the ice at +4.70 to +2.20 mm from bregma according to brain atlas [50].
The mPFC, containing the prelimbic, infralimbic, and anterior cingulate cortices, was immediately dissected from a coronal section. mPFC and hippocampal samples for western blot and ELISA were immediately frozen on liquid nitrogen and stored at −80°C until use. mPFC and hippocampal samples for real time PCR were collected in RNAlatter reagent (Applied Biosystems, Warrington, UK), and stored at −30°C until use.
These sentences have been added in the manuscript “material and method” section from:
Page#4
Line#123-129
After the RAM task, the rats were anaesthetized using isoflurane and brains were rapidly separated from the skull, and the hippocampus were bilaterally collected. Afterwards, olfactory bulbs were removed and coronal section was made on ice at +4.70 to +2.20 mm from bregma according to brain atlas [50]. The mPFC, containing the prelimbic, infralimbic, and anterior cingulate cortices, was immediately dissected from a coronal section. mPFC and hippocampal samples for Enzyme-linked immunosorbent assay (ELISA) were immediately frozen on liquid nitrogen and stored at −80°C until use.
Page#5
Line#176-181
After the Y-maze test and NOR test, the rats were anaesthetized using isoflurane and the mPFCs, hippocampus and plasma were sampled as described above. mPFC and hippocampal samples for ELISA were immediately frozen on liquid nitrogen and stored at −80°C until use. mPFCs samples for western blot were immediately frozen on liquid nitrogen and stored at −80°C until use. mPFCs samples for real time PCR were collected in RNAlatter reagent (Applied Biosystems, Warrington, UK), and stored at −30°C until use.
“PFC” in the manuscript were revised to “mPFC”.
Comment #4:
There appears to be a great deal of variance in the expression of p-CREB (Fig. 7B). Did the expression of this or other markers correlate with any behavioural measure?
Answer:
Thank you for your question. We have carried out correlation analysis, and have found significant correlation between biological markers (p-CREB, p-CaMKII and BDNF levels) and behavioural measures [as percentage of alteration in Y-maze and Discrimination Index (DI) in NOR test]. These observations have been summarized in “Supplemental table 2”.
Also, one sentence has been added in the “Discussion” section as below-
Page # 13
Line # 418-419
Moreover, the improvements of learning and memory were positively correlated with the levels of memory-related substrates ¾ p-CaMKII, p-CREB, and BDNF (supplementary file 2).
Comment # 5:
The radial arm maze should be analyzed as a repeated-measures ANOVA. As well, please indicate on Fig. 3A and 3C which datapoints significantly differ from control.
Answer:
Thank you for your suggestion. Our research group uses two-way ANOVA for the analysis of radial arm maze data for the last 15 years. We have used this because two-way ANOVA and repeated-measures ANOVA can actually mean the same thing. Two-way typically refers to two independent variables (factors) and their interaction that affect the dependent variable. While “Repeated measures” typically means that one of the factors is within-subjects. So, if one of the independent variables is within-subjects factor (e.g. block seen through three measurements of some construct taken three times) and the second one is some sort of between-groups factor [e.g. (TB and CN rats) groups] then we would apply two-way repeated measures ANOVA. That’s why we have analysed the data for Radial Arm Maze (Figure. 3) by a two-way ANOVA followed by Fisher’s PLSD post-hoc tests and found significant interactions and differences between two groups.
According to your comment, by using asterisks, we have indicated the data points, where the performances were significantly different between TB vs. Control rats (Figure 3A, 3C).
MINOR COMMENTS
Comment#6:
The diagrams in Fig. 1 and 2 have some points of confusion and could be improved for clarity.
6(a) Please, indicate how much time passed from “animal arrival” when the animals were 35 days old and “Day 0” when the animals went into protocol.
Answer: We are sorry for making ambiguous descriptions for the diagrams. Rats were maintained for 10 days prior to control or TB food administration. Therefore, “10 days” has been added in the Figures 1 and 2.
Comment#6:
6(b) The label “TB food administration” should be “Control or TB food administration”
Answer: We agree the comment. The part has been revised on Figure 1 and 2.
Comment#6:
6(c) The word “starvation” is presumed to be inaccurate and should be replaced with “fasting”
Answer: We have corrected this in Figure 1.
Comment#6:
6(d) The Methods section indicates a week of handling prior to testing; this should be indicated on the diagram.
Answer: Thank you for pointing it out. It has been indicated in Figure 1.
Comment#7:
The data that is omitted in the instances of “data not shown,” should be included in a supplemental file. These data address points of interest to this reader.
Answer: Thanks for your suggestion. We have attached the omitted data in the supplemental file 2.
Comment#8
While the manuscript is largely well-written, there are many instances of typographical errors. This is not a complete list:
a. Pg. 2, “transverses” should be “traverses”
b. Pg. 2, “medium” should be “medial”
c. Pg. 4 and throughout the manuscript, “alterations” should be “alternations”
d. Pg. 11, “trails” should be “trials”
Answer: Thank you for pointed out. These parts have been corrected in the revised manuscript. Also, we have also fixed some other mistakes e.g.
Page# 5 and line#186: γ-glutamyl transpeptidase
Page# 6 and line#224: β-actin
Page# 6 and line#235: PLSD
Page# 12 and line#364: fewer
Page# 13 and line#453: in
References:
1. Maurice, T.; Hiramatsu, M.; Itoh, J.; Kameyama, T.; Hasegawa, T.; Nabeshima, T. Behavioral evidence for a modulating role of sigma ligands in memory processes. I. Attenuation of dizocilpine (MK-801)-induced amnesia. Brain Res. 1994, 647, 44–56, doi:10.1016/0006-8993(94)91397-8.
2. Kameyama, T.; Ukai, M.; Shinkai, N. Ameliorative effects of tachykinins on scopolamine-induced impairment of spontaneous alternation performance in mice. Methods Find. Exp. Clin. Pharmacol. 1998, 20, 555–60.
3. Ukai, M.; Shinkai, N.; Kameyama, T. Involvement of dopamine receptors in beneficial effects of tachykinins on scopolamine-induced impairment of alternation performance in mice. Eur. J. Pharmacol. 1998, 350, 39–45.
4. Ennaceur, A.; Neave, N.; Aggleton, J.P. Spontaneous object recognition and object location memory in rats: the effects of lesions in the cingulate cortices, the medial prefrontal cortex, the cingulum bundle and the fornix. Exp. brain Res. 1997, 113, 509–19, doi:10.1007/PL00005603.
5. Quillfeldt, J.A. Behavioral methods to study learning and memory in rats. In Rodent Model as Tools in Ethical Biomedical Research; Springer, 2016; pp. 271–311.
6. Graff, J.; Woldemichael, B.T.; Berchtold, D.; Dewarrat, G.; Mansuy, I.M. Dynamic histone marks in the hippocampus and cortex facilitate memory consolidation. Nat. Commun. 2012, 3, 991, doi:10.1038/ncomms1997.
7. Ferino, F.; Thierry, A.M.; Glowinski, J. Anatomical and electrophysiological evidence for a direct projection from Ammon’s horn to the medial prefrontal cortex in the rat. Exp. brain Res. 1987, 65, 421–426.
8. Laroche, S.; Davis, S.; Jay, T.M. Plasticity at hippocampal to prefrontal cortex synapses: dual roles in working memory and consolidation. Hippocampus 2000, 10, 438–446, doi:10.1002/1098-1063(2000)10:4<438::aid-hipo10>3.0.CO;2-3.
9. Thierry, A.M.; Gioanni, Y.; Degenetais, E.; Glowinski, J. Hippocampo-prefrontal cortex pathway: anatomical and electrophysiological characteristics. Hippocampus 2000, 10, 411–419, doi:10.1002/1098-1063(2000)10:4<411::aid-hipo7>3.0.CO;2-A.
10. Yoon, T.; Okada, J.; Jung, M.W.; Kim, J.J. Prefrontal cortex and hippocampus subserve different components of working memory in rats. Learn. Mem. 2008, 15, 97–105, doi:10.1101/lm.850808.
11. Khan, Z.U.; Muly, E.C. Molecular mechanisms of working memory. Behav. Brain Res. 2011, 219, 329–41, doi:10.1016/j.bbr.2010.12.039.
12. Han, S.; Bhattarai, J. Phasic and tonic type A γ-Aminobutryic acid receptor mediated effect of Withania somnifera on mice hippocampal CA1 pyramidal Neurons. J. Ayurveda Integr. Med. 2014, 5, 216, doi:10.4103/0975-9476.146541.
13. Ravichandran, V.; Kim, M.; Han, S.; Cha, Y. Stachys sieboldii Extract Supplementation Attenuates Memory Deficits by Modulating BDNF-CREB and Its Downstream Molecules, in Animal Models of Memory Impairment. Nutrients 2018, 10, 917.
14. Bontempi, B.; Laurent-Demir, C.; Destrade, C.; Jaffard, R. Time-dependent reorganization of brain circuitry underlying long-term memory storage. Nature 1999, 400, 671.
15. Mumby, D.G.; Glenn, M.J. Anterograde and retrograde memory for object discriminations and places in rats with perirhinal cortex lesions. Behav. Brain Res. 2000, 114, 119–134.
16. Ramos, J.M.J. Long-term spatial memory in rats with hippocampal lesions. Eur. J. Neurosci. 2000, 12, 3375–3384.
17. Witter, M.P.; Naber, P.A.; Van Haeften, T.; Machielsen, W.C.M.; Rombouts, S.A.R.B.; Barkhof, F.; Scheltens, P.; da Silva, F.H. Cortico-hippocampal communication by way of parallel parahippocampal-subicular pathways. Hippocampus 2000, 10, 398–410.
18. Frankland, P.W.; Bontempi, B.; Talton, L.E.; Kaczmarek, L.; Silva, A.J. The involvement of the anterior cingulate cortex in remote contextual fear memory. Science (80-. ). 2004, 304, 881–883.
19. Jung, M.W.; Baeg, E.H.; Kim, M.J.; Kim, Y.B.; Kim, J.J. Plasticity and memory in the prefrontal cortex. Rev. Neurosci. 2008, 19, 29–46.
20. Nieuwenhuis, I.L.C.; Takashima, A. The role of the ventromedial prefrontal cortex in memory consolidation. Behav. Brain Res. 2011, 218, 325–334.
21. Squire, L.R.; Alvarez, P. Retrograde amnesia and memory consolidation: a neurobiological perspective. Curr. Opin. Neurobiol. 1995, 5, 169–177.
22. Carew, T.J. Molecular enhancement of memory formation. Neuron 1996, 16, 5–8, doi:10.1016/S0896-6273(00)80016-1.
23. Martin, K.C.; Barad, M.; Kandel, E.R. Local protein synthesis and its role in synapse-specific plasticity. Curr. Opin. Neurobiol. 2000, 10, 587–92, doi:10.1016/S0959-4388(00)00128-8.
24. Kelleher III, R.J.; Govindarajan, A.; Tonegawa, S. Translational regulatory mechanisms in persistent forms of synaptic plasticity. Neuron 2004, 44, 59–73.
25. Briand, L.A.; Gritton, H.; Howe, W.M.; Young, D.A.; Sarter, M. Modulators in concert for cognition: modulator interactions in the prefrontal cortex. Prog. Neurobiol. 2007, 83, 69–91, doi:10.1016/j.pneurobio.2007.06.007.
26. Dudai, Y. Molecular bases of long-term memories: a question of persistence. Curr. Opin. Neurobiol. 2002, 12, 211–216.
27. Lamprecht, R.; LeDoux, J. Structural plasticity and memory. Nat. Rev. Neurosci. 2004, 5, 45–54, doi:10.1038/nrn1301.
28. Park, H.; Poo, M. Neurotrophin regulation of neural circuit development and function. Nat. Rev. Neurosci. 2013, 14, 7.
29. Henderson, C.E. Role of neurotrophic factors in neuronal development. Curr. Opin. Neurobiol. 1996, 6, 64–70.
30. Levi-Montalcini, R. The nerve growth factor 35 years later. Science (80-. ). 1987, 237, 1154–1162.
31. Davies, A.M. The neurotrophic hypothesis: where does it stand? Philos. Trans. R. Soc. London. Ser. B Biol. Sci. 1996, 351, 389–394.
32. Miyamoto, E. Molecular mechanism of neuronal plasticity: induction and maintenance of long-term potentiation in the hippocampus. J. Pharmacol. Sci. 2006, 100, 433–42.
33. Leinninger, G.M.; Backus, C.; Uhler, M.D.; Lentz, S.I.; Feldman, E.L. Phosphatidylinositol 3-kinase and Akt effectors mediate insulin-like growth factor-I neuroprotection in dorsal root ganglia neurons. FASEB J. Off. Publ. Fed. Am. Soc. Exp. Biol. 2004, 18, 1544–1546, doi:10.1096/fj.04-1581fje.
34. Vitolo, O. V; Sant’Angelo, A.; Costanzo, V.; Battaglia, F.; Arancio, O.; Shelanski, M. Amyloid beta -peptide inhibition of the PKA/CREB pathway and long-term potentiation: reversibility by drugs that enhance cAMP signaling. Proc. Natl. Acad. Sci. U. S. A. 2002, 99, 13217–21, doi:10.1073/pnas.172504199.
35. Zhao, L.; Brinton, R.D. Vasopressin-induced cytoplasmic and nuclear calcium signaling in embryonic cortical astrocytes: dynamics of calcium and calcium-dependent kinase translocation. J. Neurosci. 2003, 23, 4228–4239.
36. Yan, X.; Liu, J.; Ye, Z.; Huang, J.; He, F.; Xiao, W.; Hu, X.; Luo, Z. CaMKII-Mediated CREB phosphorylation is involved in Ca2+-Induced BDNF mRNA transcription and neurite outgrowth promoted by electrical stimulation. PLoS One 2016, 11, 1–22, doi:10.1371/journal.pone.0162784.
37. Kandel, E.R. The molecular biology of memory storage: a dialogue between genes and synapses. Science (80-. ). 2001, 294, 1030–1038.
38. Schulman, H.; Hanson, P.I. Multifunctional Ca2+/calmodulin-dependent protein kinase. Neurochem. Res. 1993, 18, 65–77.
39. Kennedy, M.B. Signal transduction molecules at the glutamatergic postsynaptic membrane. Brain Res. Brain Res. Rev. 1998, 26, 243–257.
40. Lisman, J.; Yasuda, R.; Raghavachari, S. Mechanisms of CaMKII action in long-term potentiation. Nat. Rev. Neurosci. 2012, 13, 169–182, doi:10.1038/nrn3192.
41. Korte, M.; Kang, H.; Bonhoeffer, T.; Schuman, E. A role for BDNF in the late-phase of hippocampal long-term potentiation. Neuropharmacology 1998, 37, 553–559.
42. Lessmann, V. Neurotrophin-dependent modulation of glutamatergic synaptic transmission in the mammalian CNS. Gen. Pharmacol. 1998, 31, 667–674.
43. Bimonte, H.A.; Nelson, M.E.; Granholm, A.-C.E. Age-related deficits as working memory load increases: relationships with growth factors. Neurobiol. Aging 2003, 24, 37–48.
44. Bimonte-Nelson, H.A.; Hunter, C.L.; Nelson, M.E.; Granholm, A.-C.E. Frontal cortex BDNF levels correlate with working memory in an animal model of Down syndrome. Behav. Brain Res. 2003, 139, 47–57.
45. Bilsland, J.G.; Wheeldon, A.; Mead, A.; Znamenskiy, P.; Almond, S.; Waters, K.A.; Thakur, M.; Beaumont, V.; Bonnert, T.P.; Heavens, R.; et al. Behavioral and neurochemical alterations in mice deficient in anaplastic lymphoma kinase suggest therapeutic potential for psychiatric indications. Neuropsychopharmacology 2008, 33, 685–700, doi:10.1038/sj.npp.1301446.
46. Clarke, J.R.; Cammarota, M.; Gruart, A.; Izquierdo, I.; Delgado-Garcia, J.M. Plastic modifications induced by object recognition memory processing. Proc. Natl. Acad. Sci. U. S. A. 2010, 107, 2652–2657, doi:10.1073/pnas.0915059107.
47. Ennaceur, A.; Delacour, J. A new one-trial test for neurobiological studies of memory in rats. 1: Behavioral data. Behav. Brain Res. 1988, 31, 47–59, doi:10.1016/0166-4328(88)90157-X.
48. Reger, M.L.; Hovda, D.A.; Giza, C.C. Ontogeny of Rat Recognition Memory measured by the novel object recognition task. Dev. Psychobiol. 2009, 51, 672–678, doi:10.1002/dev.20402.
49. Taglialatela, G.; Hogan, D.; Zhang, W.-R.; Dineley, K.T. Intermediate- and long-term recognition memory deficits in Tg2576 mice are reversed with acute calcineurin inhibition. Behav. Brain Res. 2009, 200, 95–99, doi:10.1016/j.bbr.2008.12.034.
50. Paxinos, G.; Watson, C. The Rat Brain in Stereotaxic Coordinates; Academic Press, 1998; ISBN 9780125476195.
(Reference numbers here are different in revised manuscript.)

Reviewer 2 Report
The manuscript submitted by Rafiad Islam entitled “Theobromine improves working memory by activating the CaMKII/CREB/BDNF pathway in rats”. This is a continuation of the same research group that has studied the role of Theobromine (TB), the major molecule in cacao beans, in neurological diseases. Previously, this group has reported that TB can promote BDNF expression. Rafia Islam et al reported the CaMKII/CREB/BDNF pathways may be involved in learning and working memory by orally treating rats.
The author has mapped the mechanism of TB in learning memory through well-designed research and well established research methods including Western blot, Real time PCR and Enzyme-linked immunosorbent assay. The data analysis methods are very appropriate for each set of data, and conclusions are faithfully bounded to the statistical results. Graphs are clearly labeled and formats are correctly presented. This whole manuscript is written in very professional English. Though this research project fits this journal very well, and the discovery is novel, a couple minor points should be addressed or explained before it is accepted for publication:
1. The author should provide explanation on the difference of numbers of animal in each assay. Such as in 3.1.3 Fig 4 BDNF only has five animals and 3.2.2 Fig 5 N=8, Fig 6 N=6 and Fig 7 N=6 (a justification for varying sample sizes is needed).
2. Please add a paragraph in the material and methods section to state the sample size of each tested assay to make sure the reader can follow the data.
3. Since Theobromine can be metabolized from caffeine, and both caffeine and theobromine may work through the similar pathway, it will be good if the author can compare the differences between caffeine and theobromine (optional).
Author Response
To Reviewer #2
(All revisions have been made in green colour in this response sheet.)
The manuscript submitted by Rafiad Islam entitled “Theobromine improves working memory by activating the CaMKII/CREB/BDNF pathway in rats”. This is a continuation of the same research group that has studied the role of Theobromine (TB), the major molecule in cacao beans, in neurological diseases. Previously, this group has reported that TB can promote BDNF expression. Rafia Islam et al reported the CaMKII/CREB/BDNF pathways may be involved in learning and working memory by orally treating rats.
The author has mapped the mechanism of TB in learning memory through well-designed research and well established research methods including Western blot, Real time PCR and Enzyme-linked immunosorbent assay. The data analysis methods are very appropriate for each set of data, and conclusions are faithfully bounded to the statistical results. Graphs are clearly labeled and formats are correctly presented. This whole manuscript is written in very professional English. Though this research project fits this journal very well, and the discovery is novel, a couple minor points should be addressed or explained before it is accepted for publication:
Comment#1:
The author should provide explanation on the difference of numbers of animal in each assay. Such as in 3.1.3 Fig 4 BDNF only has five animals and 3.2.2 Fig 5 N=8, Fig 6 N=6 and Fig 7 N=6 (a justification for varying sample sizes is needed).
Answer:
We are very sorry for making vague descriptions for the number of rats.
In the case of Y-maze (Figure 5), we have excluded 2 rats per group from analysis due to less than 6 arm entry [1].
In Aubele et al.’s research on NOR test three behaviours were evaluated during the NOR experiment and were categorized as ambulating (the crossing of at least 1 floor grid line within a 3-s period), rearing (a lifting of the forelimbs and sitting back upon the haunches), or remaining stationary (the animal remained unmoving at least during 3-s). These times spent on these three activities were measured separately. When animals showed lack of exploration activity, the researcher excluded those animals from the experimental analysis [2–6].
In NOR test (Figure 6), 2 rats, those were removed from Y-maze also excluded here and other 3 rats were excluded from the analysis that showed a lack of exploration activity. After the behavioural test, every 7 rats in “Experiment 1” and 4 rats in “Experiment 2” have been transcardially perfused with formaldehyde and brains were used for immunohistochemical analysis. Although we have tried to confirm the expression level of BDNF, p-CREB, p-CaMKII and up/downstream signals by immuno-staining many times, we could not obtain the consolidated/solid results in any brain regions. Therefore, we have abandoned the evaluation of immuno-staining in this time. Meanwhile, we have used 5 animals per group for ELISA of BDNF (Figure 4, “Experiment 1”) and 6 animals per group for western blot analysis of p-CERB and p-CaMKII, ELISA of BDNF (Figure 7 and 8, “Experiment 2”) and real time PCR of BDNF (Figure 8, “Experiment 2”). We sincerely hope you could accept our explanation.
Comment#2: Please add a paragraph in the material and methods section to state the sample size of each tested assay to make sure the reader can follow the data.
Answer:
Thanks for your commendation. As per your suggestion, we have added a paragraph in the material and methods section to mention the sample size of each tested assay.
The following paragraph has been added in the manuscript.
Page#6
Line#228-232
2.7. Sample size of each tested assay
In RAM task (Experiment 1) 12 animals were used per group. In Y-maze (Experiment 2) 8 animals and in NOR test (Experiment 2) 5 animals were used. For ELISA (Experiment 1) we used 5 animal from each group. For real time PCR, ELISA, western blot analysis of p-CERB and p-CaMKII (Experiment 2), we used 6 animal per group.
Comment#3: Since Theobromine can be metabolized from caffeine, and both caffeine and theobromine may work through the similar pathway, it will be good if the author can compare the differences between caffeine and theobromine (optional).
Answer:
Thank you for your reasonable suggestion. We also have an interest in the effects of caffeine on memory/cognitive functions.
Associated with this, one section and 8 references have been added in the discussion:
Page#12
Line#436-445
TB can be metabolized from caffeine, and both caffeine and TB may work through a similar pathway. Caffeine has some pharmacological effects in the central nervous system and beneficial effects on memory and/or cognitive functions in humans and rodents [7–11]. Since TB can be metabolized from caffeine, these components may work through similar mechanisms. For instance, caffeine and TB can pass through BBB and block cell surface adenosine receptors which distributed widely throughout cortical regions [12]. TB and caffeine also act as PDE inhibitor that increases in intracellular cAMP level [12–14]. Moreover, these reagents are known to affect Ca2+ release from intracellular stores of the brain [12]. Although, exact differences between caffeine and TB on memory and/or cognitive functions have not been examined in this study, a comparison study for these reagents may be required in the future.
References:
1. Wietrzych, M.; Meziane, H.; Sutter, A.; Ghyselinck, N.; Chapman, P.F.; Chambon, P.; Krezel, W. Working memory deficits in retinoid X receptor gamma-deficient mice. Learn. Mem. 2005, 12, 318–326, doi:10.1101/lm.89805.
2. Bilsland, J.G.; Wheeldon, A.; Mead, A.; Znamenskiy, P.; Almond, S.; Waters, K.A.; Thakur, M.; Beaumont, V.; Bonnert, T.P.; Heavens, R.; et al. Behavioral and neurochemical alterations in mice deficient in anaplastic lymphoma kinase suggest therapeutic potential for psychiatric indications. Neuropsychopharmacology 2008, 33, 685–700, doi:10.1038/sj.npp.1301446.
3. Clarke, J.R.; Cammarota, M.; Gruart, A.; Izquierdo, I.; Delgado-Garcia, J.M. Plastic modifications induced by object recognition memory processing. Proc. Natl. Acad. Sci. U. S. A. 2010, 107, 2652–2657, doi:10.1073/pnas.0915059107.
4. Ennaceur, A.; Delacour, J. A new one-trial test for neurobiological studies of memory in rats. 1: Behavioral data. Behav. Brain Res. 1988, 31, 47–59, doi:10.1016/0166-4328(88)90157-X.
5. Reger, M.L.; Hovda, D.A.; Giza, C.C. Ontogeny of Rat Recognition Memory measured by the novel object recognition task. Dev. Psychobiol. 2009, 51, 672–678, doi:10.1002/dev.20402.
6. Taglialatela, G.; Hogan, D.; Zhang, W.-R.; Dineley, K.T. Intermediate- and long-term recognition memory deficits in Tg2576 mice are reversed with acute calcineurin inhibition. Behav. Brain Res. 2009, 200, 95–99, doi:10.1016/j.bbr.2008.12.034.
7. Shabir, A.; Hooton, A.; Tallis, J.; F Higgins, M. The Influence of Caffeine Expectancies on Sport, Exercise, and Cognitive Performance. Nutrients 2018, 10, doi:10.3390/nu10101528.
8. Ritchie, K.; Carriere, I.; de Mendonca, A.; Portet, F.; Dartigues, J.F.; Rouaud, O.; Barberger-Gateau, P.; Ancelin, M.L. The neuroprotective effects of caffeine: a prospective population study (the Three City Study). Neurology 2007, 69, 536–545, doi:10.1212/01.wnl.0000266670.35219.0c.
9. van Gelder, B.M.; Buijsse, B.; Tijhuis, M.; Kalmijn, S.; Giampaoli, S.; Nissinen, A.; Kromhout, D. Coffee consumption is inversely associated with cognitive decline in elderly European men: the FINE Study. Eur. J. Clin. Nutr. 2007, 61, 226–232, doi:10.1038/sj.ejcn.1602495.
10. Santos, C.; Lunet, N.; Azevedo, A.; de Mendonca, A.; Ritchie, K.; Barros, H. Caffeine intake is associated with a lower risk of cognitive decline: a cohort study from Portugal. J. Alzheimers. Dis. 2010, 20 Suppl 1, S175-85, doi:10.3233/JAD-2010-091303.
11. Alzoubi, K.H.; Mhaidat, N.M.; Obaid, E.A.; Khabour, O.F. Caffeine Prevents Memory Impairment Induced by Hyperhomocysteinemia. J. Mol. Neurosci. 2018, 66, 222–228, doi:10.1007/s12031-018-1158-3.
12. Chen, X.; Ghribi, O.; Geiger, J.D. Caffeine protects against disruptions of the blood-brain barrier in animal models of Alzheimer’s and Parkinson’s diseases. J. Alzheimers. Dis. 2010, 20 Suppl 1, S127-41, doi:10.3233/JAD-2010-1376.
13. Yoneda, M.; Sugimoto, N.; Katakura, M.; Matsuzaki, K.; Tanigami, H.; Yachie, A.; Ohno-Shosaku, T.; Shido, O. Theobromine up-regulates cerebral brain-derived neurotrophic factor and facilitates motor learning in mice. J. Nutr. Biochem. 2017, 39, 110–116, doi:10.1016/j.jnutbio.2016.10.002.
14. Sugimoto, N.; Katakura, M.; Matsuzaki, K.; Sumiyoshi, E.; Yachie, A.; Shido, O. Chronic administration of theobromine inhibits mTOR signal in rats. Basic Clin. Pharmacol. Toxicol. 2018, 00, 1–7, doi:10.1111/bcpt.13175.
(Reference numbers here are different in revised manuscript.)

Round 2
Reviewer 1 Report
The manuscript by Islam and colleagues is improved from its initial submission. While it would have been most satisfactory for authors to have included additional observations to increase the low n on NOR test and potentially address the low novel object exploration in the control group, the present data are significant and the authors acknowledge this shortcoming in the manuscript text.
Upon explanation, it appears that the authors have analyzed their RAM data properly, but have used a different nomenclature than the reviewer. The reviewer still believes Repeated Measures ANOVA to be the proper terminology that should be used herein, but acknowledges that this is of minor concern.